# PlatyphyllenoneExerts Anti-Metastatic Effects on Human Oral Cancer Cells by Modulating Cathepsin L Expression, MAPK Pathway and Epithelial–Mesenchymal Transition

**DOI:** 10.3390/ijms22095012

**Published:** 2021-05-09

**Authors:** V. Bharath Kumar, Jen-Tsun Lin, B. Mahalakshmi, Yi-Ching Chuang, Hsin-Yu Ho, Chia-Chieh Lin, Yu-Sheng Lo, Ming-Ju Hsieh, Mu-Kuan Chen

**Affiliations:** 1Department of Medical Laboratory Science and Biotechnology, Asia University, Taichung 413, Taiwan; bharathvel@gmail.com; 2Division of Hematology and Oncology, Department of Medicine, Changhua Christian Hospital, Changhua 500, Taiwan; 111227@cch.org.tw; 3School of Medicine, Chung Shan Medical University, Taichung 40201, Taiwan; 4Post Baccalaureate Medicine, National Chung Hsing University, Taichung 402, Taiwan; 5Department of Research and Development, Vels Publisher, Bodinayakanur 625513, India; mahalakshmibharath05@gmail.com; 6Oral Cancer Research Center, Changhua Christian Hospital, Changhua 500, Taiwan; 177267@cch.org.tw (Y.-C.C.); 183581@cch.org.tw (H.-Y.H.); 181327@cch.org.tw (C.-C.L.); 165304@cch.org.tw (Y.-S.L.); 7Institute of Medicine, Chung Shan Medical University, Taichung 402, Taiwan; 8Graduate Institute of Biomedical Sciences, China Medical University, Taichung 404, Taiwan; 9Department of Otorhinolaryngology, Head and Neck Surgery, Changhua Christian Hospital, Changhua 500, Taiwan

**Keywords:** platyphyllenone, migration, invasion, oral

## Abstract

Advanced-stage oral cancers with lymph node metastasis are associated with poor prognosis and a high mortality rate. Although recent advancement in cancer treatment has effectively improved the oral cancer prognosis, the majority of therapeutic interventions are highly expensive and are associated with severe sideeffects. In the present study, we studied the efficacy of a diarylheptanoid derivative, platyphyllenone, in modulating the metastatic potential of human oral cancer cells. Specifically, we treated the human oral cancer cells (FaDu, Ca9-22, and HSC3) with different concentrations of platyphyllenone and measured the cell proliferation, migration, and invasion. The study findings revealed that platyphyllenonesignificantly inhibited the motility, migration, and invasion of human oral cancer cells. Mechanistically, platyphyllenone reduced p38 phosphorylation, decreased β-catenin and Slug, increased E-cadherin expression, and reduced cathepsin L expression, which collectively led to a reduction in cancer cell migration and invasion. Taken together, our study indicates that platyphyllenone exerts significant anti-metastatic effects on oral cancer cells by modulating cathepsin L expression, the MAPK signaling pathway, and the epithelial–mesenchymal transition process.

## 1. Introduction

Oral cancer, which is the 11th most common cancer globally, primarily occurs in the lips, oral cavity, nasopharynx, and pharynx. About 90% of all oral cancers are squamous cell carcinomas, which mainly occur due to excessive smoking and alcohol consumption [1]. Although treatment of early-stage oral cancers can be effectively carried out by surgical resection, along with post-operative chemotherapy and/or radiation therapy, advanced-stage oral cancers with lymph node metastasis can be lethal and are associated with poor prognosis [2,3]. Moreover, oral cancer treatments are highly expensive and are often associated with severe sideeffects [1].

Aberrant activation of MAPK signaling due to genetic alterations is one of the major etiologies of oral cancer. Important cellular components of the MAPK signaling pathway, including ERK1/2, JNK, and p38, play vital roles in regulating many physiological processes, such as cell proliferation, apoptosis, angiogenesis, and migration [4]. Given the significant involvement of MAPK signaling in neoplastic transformation, severe small molecular inhibitors targeting specific components of the pathway have been identified, and the majority of the inhibitors have shown promising anti-cancer effects in clinical trials [5,6,7]. In case of oral squamous cell carcinoma, certain bioactive plant compounds, such as epigallocatechin-3-gallate, S-allylcysteine, pterostilbene, and resveratrol, have been shown to inhibit cancer cell proliferation, induce apoptosis, and inhibit migration by modulating the MAPK signaling pathway [8,9,10,11].

Diarylheptanoids derived from different species of *Alpinia*, *Zingiber*, *Curcuma*, and *Alnus* are bioactive natural compounds with potential therapeutic benefits [12]. Diarylheptanoid derivatives are known to inhibit cancer cell proliferation by inducing free radical production and modulating the MAPK signaling pathway [13]. Linear diarylheptanoids such as curcuminoids are known to potentiate the anti-cancer effects of chemotherapeutics, such as cisplatin [14]. Interestingly, diarylheptanoids possess both estrogen-activating and -inhibiting effects, and in vitro/in vivo studies have shown that these compounds prevent bone loss and reduce the risk of breast and uterine cancers [15]. Moreover, these compounds have strong anti-inflammatory and anti-pathogenic effects, and thereby, are effective in treating various inflammatory and infectious diseases [16,17]. Among the various diarylheptanoids, platyphyllenone has been shown to have anti-proliferative effects in many cancer types, including pancreatic cancer and breast cancer [13,18]. In breast cancer, platyphyllenone has been shown to inhibit cancer cell proliferation by inducing the production of reactive oxygen species (ROS) and increasing the cellular levels of phospho-JNK and phospho-p38 [13]. In pancreatic cancer, platyphyllenone has been found to inhibit cancer cell proliferation and induce cell cycle arrest by inhibiting the shh-Gli-FoxM1 pathway [18].

Given the significant therapeutic benefits of platyphyllenone, we, in the present study, aimed at deciphering the effects of platyphyllenone treatment in modulating the migration and invasion of oral cancer cells. We used human oral squamous carcinoma cell lines, FaDu, Ca9-22, and HSC3, as experimental models to investigate the mode of action of platyphyllenone.

## 2. Results

### 2.1. Platyphyllenone Does Not Affect the Viability of Human Oral Cancer Cells

To check the cytotoxic effect of platyphyllenone on human oral cancer cells, FaDu, Ca9-22, and HSC3 cell lines were treated with 2.5, 5, and 10 μM of platyphyllenone for 24 h. As observed in Figure 1B–D, none of the doses of platyphyllenone affected the viability of human oral cancer cells, indicating no cytotoxic effects of platyphyllenone on oral cancer.

### 2.2. Platyphyllenone Inhibits the Motility of Human Oral Cancer Cells

Because platyphyllenone did not exhibit any anti-proliferative effect, we next thought of investigating the anti-migratory effects of platyphyllenone on human oral cancer cells using wound closer assay. Upon treatment of FaDu, Ca9-22, and HSC3 cell lines with 2.5, 5, and 10 μM of platyphyllenone for 3, 6, and 24 h, we observed that platyphyllenone significantly inhibited the motility of oral cancer cells in a dose-dependent manner (Figure 2A–C and Appendix A). The effect of platyphyllenone was more pronounced atthe 6 h time point for FaDu and Ca9-22 cells. However, for HSC3 cells, notable effects were observed at both 3 h and 6 h time points. These findings indicate that platyphyllenoneis capable of inhibiting the motility of human oral cancer cells.

### 2.3. Platyphyllenone Inhibits Migration and Invasion of Human Oral Cancer Cells

Because platyphyllenone significantly affected the oral cancer cell motility, we next investigated the anti-metastatic effect of platyphyllenone using a transwell assay. As observed in Figure 3A,B, the treatment with different doses of platyphyllenone significantly inhibited the migration of FaDu, Ca9-22, and HSC3 cells in a dose-dependent manner. Similarly, platyphyllenone was found to inhibit the invasion of oral cancer cells in a dose-dependent manner (Figure 3C,D). These findings further clarify the anti-metastatic effect of platyphyllenone on oral cancer cells.

### 2.4. Platyphyllenone Reduces Motility of Human Oral Cancer Cells by Modulating p38 Pathway

Next, we thought of investigating the mode of action of platyphyllenone in inhibiting the metastasis of oral cancer. Because the MAPK pathway components are frequently overexpressed in many cancer types and play crucial roles in regulating cell migration, we were particularly concerned with investigating the expressions and activity of specific MAPK pathway components (ERK1/2, p38, and JNK1/2). As reported in Figure 4A–D, the treatment with platyphyllenone significantly reduced the phosphorylation of p38 in FaDu and Ca9-22 cells. Because platyphyllenone significantly suppressed p38 phosphorylation, we further investigated the platyphyllenoneand p38 crosstalk. Specifically, we co-treated FaDu and Ca9-22 cells with platyphyllenone and a specific p38 pathway inhibitor (SB203580) and checked the cell motility. As reported in Figure 5 and Appendix A, platyphyllenone and SB203580 cotreatment further reduced the motility of oral cancer cells compared to platyphyllenone treatment alone. This reduction was statistically significant. These findings indicate that platyphyllenone exerts its anti-migratory activity by modulating the p38 signaling pathway.

### 2.5. Platyphyllenonealters Expression of Epithelial–Mesenchymal Transition Proteins

To further confirm the mode of action of platyphyllenone in modulating metastasis, we cotreated oral cancer cells with platyphyllenone and SB203580. Similar to cell motility, the cotreatment further reduced the oral cancer cell migration compared to platyphyllenone treatment alone (Figure 6A,B). Next, we investigated whether platyphyllenone modulates the epithelial–mesenchymal transition, which is a characteristic feature of the majority of metastatic cells. Specifically, we treated the cells with platyphyllenone and checked the expression of epithelial–mesenchymal transition proteins, including β-catenin, Slug, and E-cadherin. As observed in Figure 6C,D, platyphyllenone significantly reduced the expression of β-catenin and Slug, and increased the expression of E-cadherin in human oral cancer cells. These findings indicate that platyphyllenone helps maintain the epithelial phenotype in human cancer cells by increasing the expression of E-cadherin, which in turn reduces the ability of cancer cells to migrate.

### 2.6. Platyphyllenone Alters the Expression of Cathepsin L in Human Oral Cancer Cells

Next, we investigated whether platyphyllenone has any effect of cathepsins, which are cellular proteases involved in neoplastic transformation, cancer progression, and metastasis. Specifically, we assessed the expressions of cathepsin L (CTSL), cathepsin S (CTSS), cathepsin A (CTSA), and cathepsin Z (CTSZ) in platyphyllenone-treated oral cancer cells. As reported in Figure 7A,B, platyphyllenone treatment significantly reduced the pression of cathepsin L; however, no effect of platyphyllenone was observed on the expression levels of cathepsins S, A, and Z. To further analyze the effect of cathepsin L in platyphyllenone-mediated migration inhibition, we subsequently overexpressed cathepsin L in oral cancer cells and treated the cells with platyphyllenone. The expression of cathepsin L in FaDu and Ca9-22 cell lines after overexpression is presented in Figure 7C,D. By conducting transwell migration and invasion assays using cathepsin L-overexpressed, platyphyllenone-treated cells, we observed that overexpression of cathepsin L significantly increased the migration and invasion of oral cancer cells, and that the anti-migratory/anti-invasive effects of platyphyllenonewere also diminished by cathepsin L overexpression (Figure 7E,F). These findings clearly indicate that platyphyllenone mediates anti-metastatic effects by reducing the expression of cathepsin L.

## 3. Discussion

In the present study, we evaluated the effects of platyphyllenone, a plant-derived diarylheptanoid derivative, in modulating the proliferation, migration, and invasion of human oral cancer cells. Although no anti-proliferative effect was observed (Figure 1), platyphyllenone was found to significantly reduce the motility, migration, and invasion of human oral cancer cells in a dose-dependent manner.

Previous studies investigating the anti-cancer effect of platyphyllenone have shown that the compound is capable of inhibiting the proliferation of human breast and lung cancer cells [13,19]. However, in these previous studies, very high concentrations of platyphyllenone (25 µM and 50 µM) were used to investigateits cytotoxic effects. In the present study, much lower concentrations of platyphyllenone were used; for this reason, it is probable that no cytotoxic effect was observed. Moreover, mechanistically, platyphyllenone has been shown to induce apoptosis and cell cycle arrest by triggering ROS production and the MAPK signaling pathway. In contrast, we observed that platyphyllenone downregulated the MAPK pathway components. Taken together, these findings suggest that distinct mode of action of platyphyllenone exists for different cancer types, and that the anti-cancer effect of platyphyllenone is cell type specific [20].

Regarding cancer metastasis, we observed that platyphyllenone significantly inhibited the migration and invasion of oral cancer cells by downregulating the phosphorylation of specific components of MAPK pathway, including p38 and JNK1/2 (Figure 2, Figure 3 and Figure 4). Of these components, the most pronounced effect was observed for p38; thus, we further validated the involvement of platyphyllenone–p38 crosstalk in our system. As expected, we observed that the cotreatment with platyphyllenone and a specific p38 pathway inhibitor further reduced oral cancer cell migration compared to the platyphyllenone treatment alone (Figure 5 and Figure 6A,B).

Previous studies deciphering the anti-metastatic effects of phytochemicals have shown sodium danshensu significantly inhibits the migration and invasion of human oral cancer cells by downregulating the phosphorylation of p38 [21]. Similarly, other bioactive plant compounds, such as Kahweol acetate and desacetylnimbinene, have been shown to inhibit cancer metastasis by reducing the phosphorylation of p38 [22,23]. All these observations are in line with our findings, which justify the significant involvement of p38 signaling pathway in mediating anti-metastatic effects of platyphyllenone [24]. 

Regarding epithelial–mesenchymal transition, we observed that platyphyllenone significantly reduced the expression of β-catenin and Slug and increased the expression of E-cadherin in human oral cancer cells (Figure 6C,D). The proteins involved in epithelial–mesenchymal transition have emerged as a potential therapeutic target because of their pivotal roles in increasing cancer metastasis and causing chemoresistance. According to our findings, platyphyllenoneis capable of maintaining the epithelial phenotype of cancer cells by increasing the expression of E-cadherin. These findings justify that platyphyllenone suppresses oral cancer metastasis by preventing epithelial to mesenchymal transition. In line with the present study’s findings, previous studies have shown that curcumin can exert potential anti-metastatic effects in many cancer types by inhibiting the epithelial–mesenchymal transition process [25]. In terms of the chemical similarity between curcumin (a known PAINS compound) and platyphyllenone, platyphyllenone has a less cytotoxic effect compared with curcumin. Similarly, other phytochemicals, such as propolin C and *Duchesnea indica*, have been shown to inhibit lung cancer metastasis by downregulating the expressions of mesenchymal markers (N-cadherin, vimentin, snail, fibronectin, and Slug), and upregulating the expression of epithelial marker (E-cadherin) [26,27].

Our experiments on cathepsins showed that platyphyllenone significantly reduced the expression of cathepsin L, and that overexpression of cathepsin L significantly diminished the anti-metastatic effects of platyphyllenone (Figure 7). These findings signify the involvement of cathepsin L in platyphyllenone-mediated anti-metastatic process. In this context, previous studies have shown that increased expression and nuclear translocation of cathepsin L is associated with induction of epithelial–mesenchymal transition, which in turn can increase cancer metastasis and cause drug resistance [28,29]. Thus, our findings suggest a potential role of platyphyllenone in modulating cathepsin L-induced epithelial–mesenchymal transition in human oral cancer, which may have vital therapeutic implications in managing metastatic carcinomas. Further studies are required to firmly establish how platyphyllenone-mediated reduction in cathepsin L expression may affect the epithelial to mesenchymal transition and control cancer metastasis.

## 4. Materials and Methods

### 4.1. Cell Culture

The experiments were conducted using human tongue squamous carcinoma cell lines, FaDu (ATCC, Manas, VA, USA), Ca9-22, and HSC3 (JCRB, Shinjuku, Japan). Fetal bovine serum (10%) supplemented Eagle’s Minimum Essential Medium (MEM, Life Technologies, Grand Island, NY, USA) and Dulbecco’s Modified Eagle Medium (DMEM, Life Technologies, Grand Island, NY, USA) were used to culture HSC3 cells and FaDu and Ca9-22 cells, respectively. A humidified incubator was used to culture the cells (pH 7.4; temp: 37 °C; CO_2_: 5%).

### 4.2. Compound

Platyphyllenone (Figure 1A) with ≥98% purity was purchased from ChemFaces (CheCheng Rd, Wuhan, PRC). The stock solution (100 mM) of platyphyllenonewas prepared in dimethyl sulfoxide (DMSO); for the treatment, the final DMSO concentration was kept at <0.2%. Antibodies and the specific p38pathway inhibitor (SB203580) were procured from Cell Signaling Technology, Inc. (Danvers, MA, USA) and Santa Cruz Biotechnology (Santa Cruz, CA, USA), respectively.

### 4.3. MTT Assay

Initially, the cells cultured in 24-well plates were incubated with 2.5, 5, and 10 µM of platyphyllenone for 24 h (pH 7.4). The cells without any treatment were used as control. Finally, the cell viability was determined using an MTT (3-(4,5-dimethylthiazol-2-yl)-2,5-diphenyltetrazolium bromide) assay.

### 4.4. In Vitro Wound Closure

The cells were cultured in culture-insert wells (ibidi, Martinsried, Germany) overnight for self-insertion. After wound formation, the cells were treated with different concentrations of platyphyllenone (2.5, 5, and 10 μM) for 0, 3, 6, and 24 h (pH 7.4). Finally, the photographs of wound closure were obtained, and the mean crawling distance was determined.

### 4.5. Cell Migration and Invasion Assay

Human oral cancer cell migration and invasion was determined using a transwell assay (Greiner Bio-One, NC, USA). Briefly, the cells cultured in transwell inserts (the upper well containing serum-free medium) were treated with different concentrations of platyphyllenone (2.5, 5 and 10 μM) for 24 h (pH 7.4). Next, photographs of transwell inserts were obtained using a light microscope.

### 4.6. Western Blot Assay

The cells were lysed using lysis buffer to extract protein samples, which were subsequently subjected to electrophoretic separation using 10% polyacrylamide gel. The protein bands were then transferred onto polyvinylidene fluoride (PVDF) membranes (Millipore Corporation, Milford, MA, USA). The blocking of membranes was done for 1 h using 5% non-fat milk (TBST), followed by incubation with indicated primary antibodies for 24 h (4 °C). Next, the incubation with corresponding peroxidase-conjugated secondary antibodies was carried out at room temperature, and the bands were visualized using ECL detection system.

### 4.7. Transfection Assay

The pCMV3/Cathepsin L (CTSL) was synthesized at Sino Biological (Shanghai, China). The plasmid transfections were performed for 24 h using TurboFect Transfection Reagent (Thermo Scientific, Waltham, MA, USA).

### 4.8. Statistical Analysis

The data collected from three independent experiments are represented as mean ± SD. All statistical analyses were carried out using Student’s *t*-test (SigmaPlot 10.0, San Jose, CA, USA). A *p* value of <0.05 was considered as statistically significant.

## 5. Conclusions

The study findings reveal that platyphyllenone has potent anti-migratory and anti-invasive effects on human oral cancer cells. Mechanistically, platyphyllenone exerts its anti-cancer effects by reducing the phosphorylation of p38, upregulating the expression of E-cadherin, and downregulating the expression of cathepsin L. Taken together, the study identifies platyphyllenone as a potential anti-metastatic agent that may be used clinically to manage highly metastatic human oral squamous cell carcinomas.

## Figures and Tables

**Figure 1 ijms-22-05012-f001:**
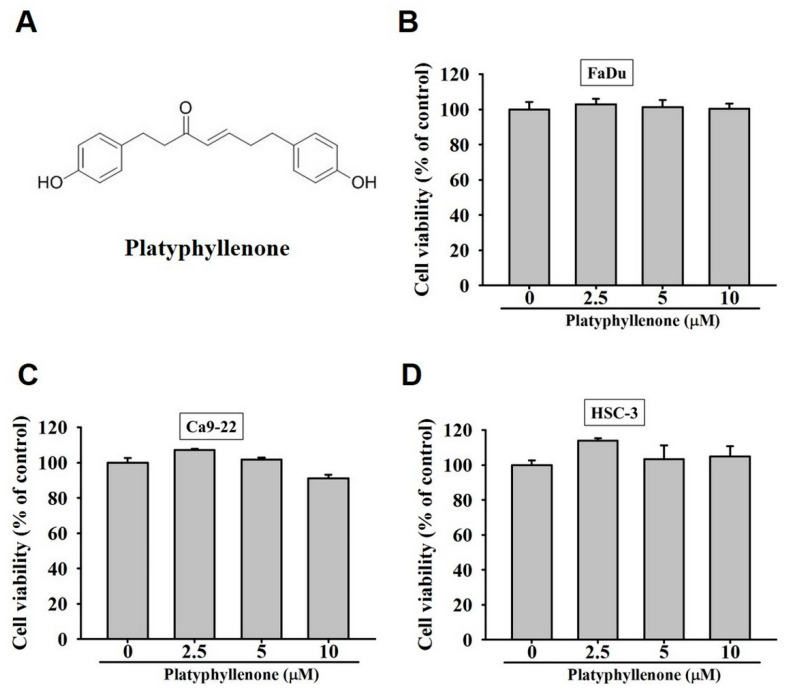
Cytotoxicity of platyphyllenoneon human oral cancer cells. (**A**) The chemical structure of platyphyllenone. (**B**) FaDu. (**C**) Ca9-22, and (**D**) HSC3 cell lines were incubated with 0, 2.5, 5, and 10 μM of platyphyllenone for 24 h, and MTT assay was conducted to determine the cell viability. The values obtained from three independent experiments are denoted as mean ± SD.

**Figure 2 ijms-22-05012-f002:**
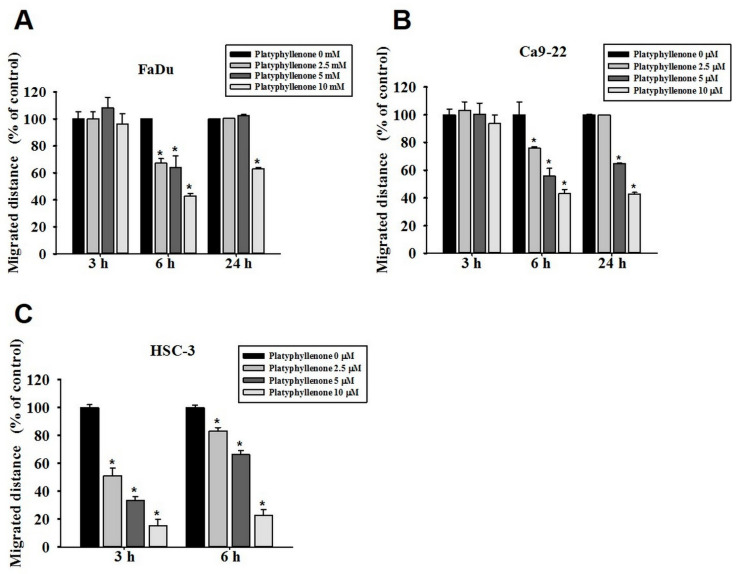
Platyphyllenone inhibits motility ofhuman oral cancer cells. Wound closer assay was conducted to assess the human oral cancer cell motility inplatyphyllenone-treated (**A**) FaDu, (**B**) Ca9-22 and (**C**) HSC3 cells. The values obtained from three independent experiments are denoted as mean ± SD. * *p* < 0.05, compared to the control (no treatment) group.

**Figure 3 ijms-22-05012-f003:**
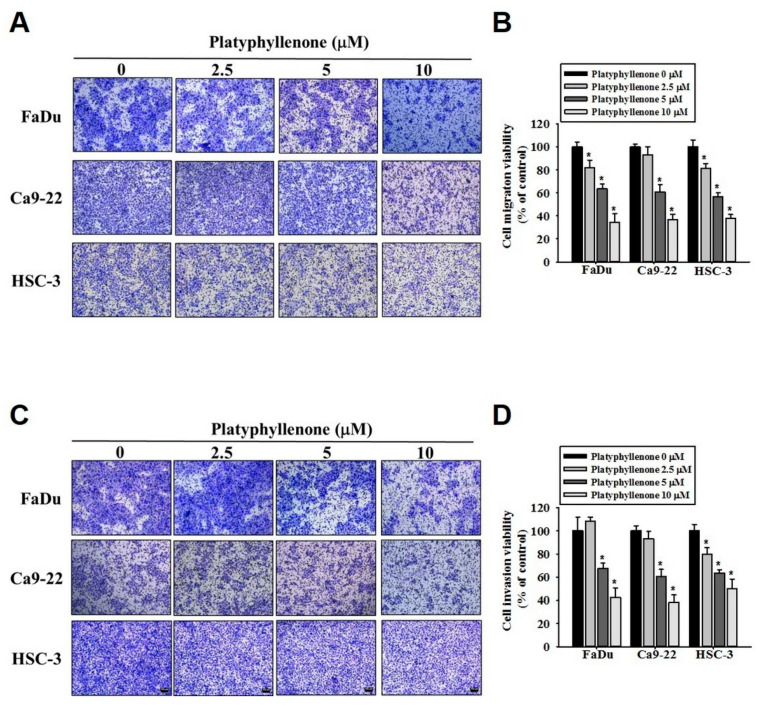
Platyphyllenone prevents human oral cancer metastasis. The effect of platyphyllenone on migration (**A**) and invasion (**C**) of FaDu, Ca9-22 and HSC3 cells was determined using a transwell assay. The cell percentages in migration (**B**) and invasion (**D**) assays are represented as bar diagrams. The values obtained from three independent experiments are denoted as mean ± SD. * *p* < 0.05, compared to the control (no treatment) group.

**Figure 4 ijms-22-05012-f004:**
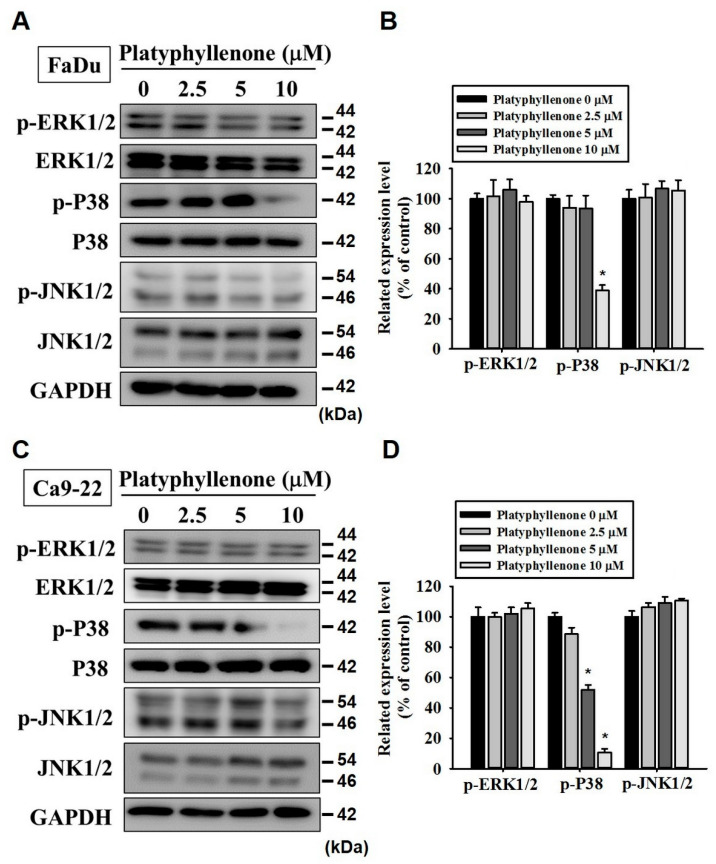
Platyphyllenone inhibits p38 and JNK1/2 pathways in human oral cancer cells. The phosphorylated and total protein levels of ERK1/2, p38, and JNK were measured in platyphyllenone-treated FaDu and Ca9-22 cell lines using Western blot analysis (**A**,**C**). The values obtained from three independent experiments are denoted as mean ± SD (**B**,**D**). * *p* < 0.05, compared to the control group.

**Figure 5 ijms-22-05012-f005:**
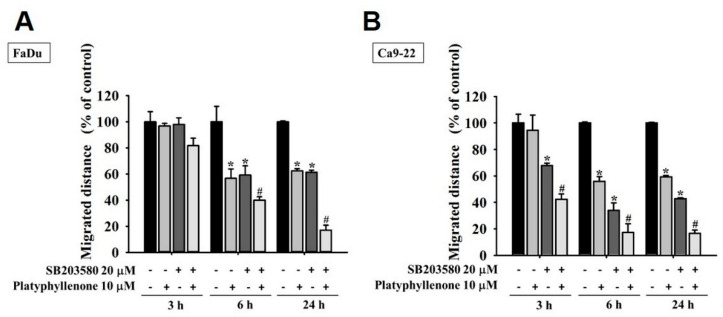
Effect of platyphyllenone and SB203580 co-treatment on oral cancer cell motility. Wound closer assay was conducted to assess the cell motility in platyphyllenone and SB203580-cotreated FaDu (**A**) and Ca9-22 cells (**B**). The values obtained from three independent experiments are denoted as mean ± SD. * *p* < 0.05, compared to the control group; # *p* < 0.05, compared to the platyphyllenone-treated group.

**Figure 6 ijms-22-05012-f006:**
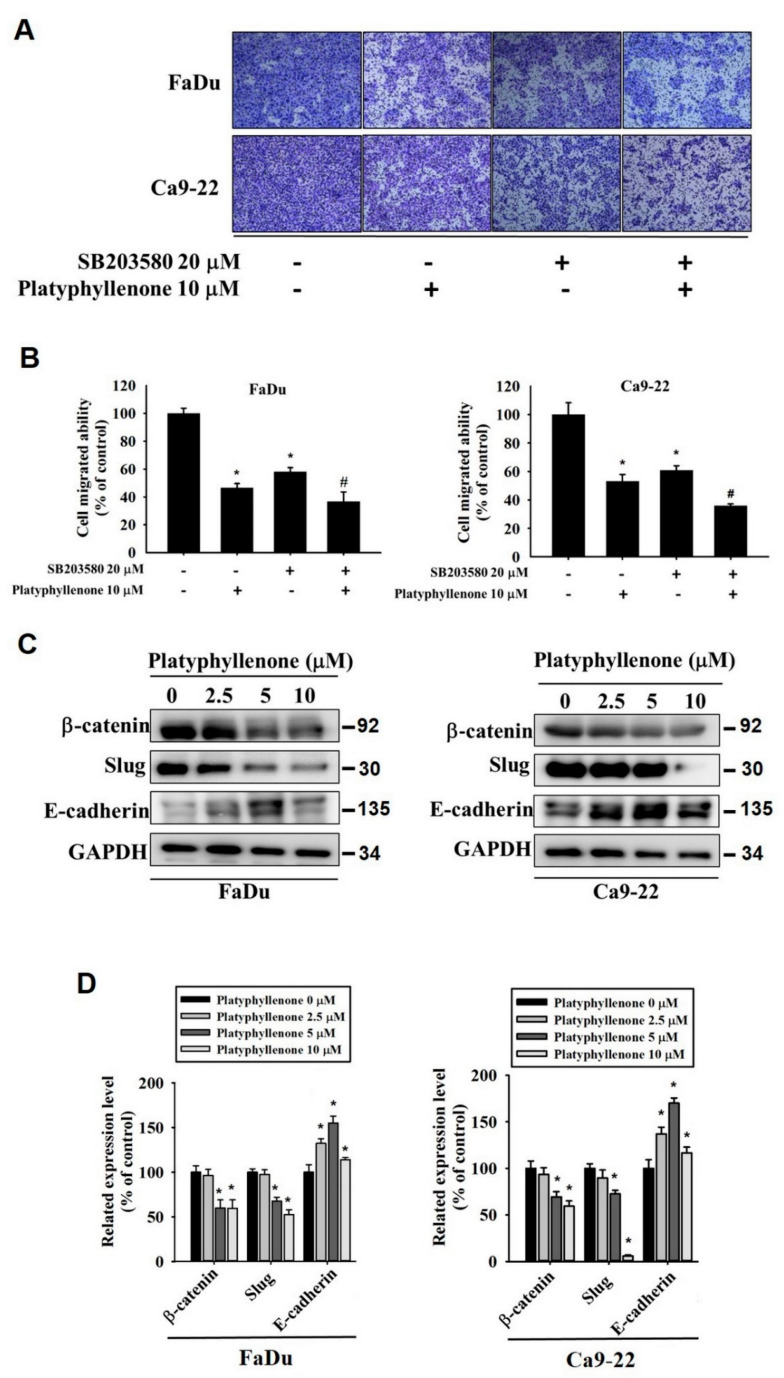
Effect of platyphyllenone and SB203580 co-treatment on cell migration and epithelial–mesenchymal transition proteins in human oral cancer cells. A transwell assay was conducted to assess the cell migration (**A**,**B**) in platyphyllenone and SB203580-cotreated FaDu and Ca9-22 cells. Platyphyllenone treatment-induced changes in protein expressions of β-catenin, slug and E-cadherin were assessed in FaDu and Ca9-22 cell lines using Western blot analysis (**C**,**D**). The values obtained from three independent experiments are denoted as mean ± SD. * *p* < 0.05, compared to the control group; # *p* < 0.05, compared to the platyphyllenone-treated group.

**Figure 7 ijms-22-05012-f007:**
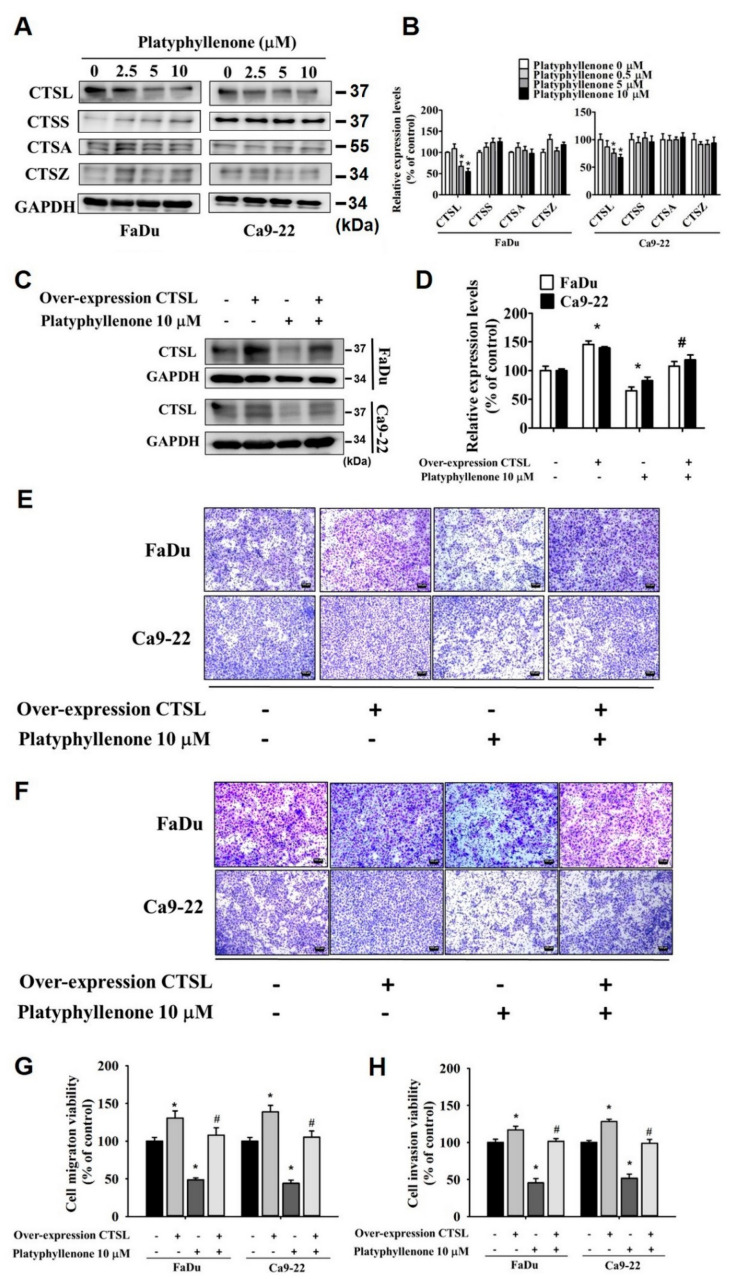
Effect of cathepsin L (CTSL) on platyphyllenone-inhibited migration and invasion of human oral cancer cells. Platyphyllenone treatment-induced changes in protein expressions of cathepsin L (CTSL), cathepsin S (CTSS), cathepsin A (CTSA) and cathepsin Z (CTSZ) were assessed in FaDu and Ca9-22 cell lines using Western blot analysis (**A**,**B**). The expression of cathepsin L after being transfected with cathepsin L plasmids in FaDu and Ca9-22 cell lines (**C**,**D**). The cell migration (**E**,**G**) was assessed using a transwell assay after being transfected with cathepsin L plasmids for 24 h, andwas then subjected to treatment with 24 h of platyphyllenone in FaDu and Ca9-22 cell lines. The cell invasion (**F**,**H**) was assessed using a transwell assay after being transfected with cathepsin L plasmids for 24 h, andwas then subjected to treatment with 24 h of platyphyllenone in FaDu and Ca9-22 cell lines. The values obtained from three independent experiments are denoted as mean ± SD. * *p* < 0.05, compared to the control group; # *p* < 0.05, compared to the platyphyllenone-treated group.

## Data Availability

This study did not report any data.

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
