# Peer review of "PlatyphyllenoneExerts Anti-Metastatic Effects on Human Oral Cancer Cells by Modulating Cathepsin L Expression, MAPK Pathway and Epithelial–Mesenchymal Transition"

_ijms, 2021, doi:10.3390/ijms22095012_

Round 1

Reviewer 1 Report

The manuscript entitled "Platyphyllenone exerts anti-metastatic effects on human oral cancer cells by modulating cathepsin L expression, MAPK pathway and epithelial-mesenchymal transition" (Velmurugan et al) reports a detailed biological evaluation of the anti-metastatic properties of platyphyllenone on oral cancer cells (i.e. FaDu, Ca9-22 and HSC-3).

The manuscript is well written and easy to follow. The reported results are interesting and the adopted experimental protocols are well described and scientifically sound.

The following observation has been raised:

  1. The chemical structure of platyphyllenone should be included
  2. Panels A, C, E in figure 2 and panels A and B in figure 5 appear to be rduntant and should be included in the supplementary information.
  3. Please replace “As observed in figure” (lines 127, 133 and 176 ) with “As reported in figure”
  4. Please for sake of clarity resentence “However, platyphyllenone and SP600125 cotreatment didn’t observe the same result (data not shown)” (lines 135-136). Furthermore the results should be reported in the manuscript
  5. Given the chemical similarity between curcumin (a known PAINS compound) and platyphyllenone the authors should comment on that aspect.
  6. As platyphyllenone is an acid compound and can be ionized, the authors should specify the pH value at which the different tests were carried out.

Reviewer 2 Report

The manuscript entitled "Platyphyllenone exerts anti-metastatic effects on human oral cancer cells by modulating cathepsin L expression, MAPK pathway and epithelial-mesenchymal transition" by Velmurugan et al. deals with a very hot topic - advanced stage oral cancers. Their study focuses on the activity of platyphyllenone towards oral squamous carcinomas - FaDu, Ca9-22 and HSC3 - a drug active towards lung and breast cancers. Authors found that the investigated compound does not exhibit antiproliferative activity, however it exerts antimetastatic effect on the studied cell lines through modulation of the expression of L-cathepsin, MAPK pathway and epithelial-mesenchymal transition.
I find the research plan being correctly planned and presented. Despite the poor antiproliferative activity of the compounds in the tested range of concentrations Authors decided to perform more detailed study on the mechanism of action in cells. Anyway, I wonder if the antiproliferative effect would be visible in the higher concentrations of drug as in case of lung and breast cancers? I think that the time range in motility studies 6 h vs. 24 h is quite long, since the observed changes in most cases are significant. Did Authors do more pictures between these time intervals?
I think that the introduction lacks the information concerning platyphyllenone - what is the structure? Why did the Authors decide to check its activity towards oral cancers?
line 136 did not instead of "didn't"

Author Response

Reviewer 2

The manuscript entitled "Platyphyllenone exerts anti-metastatic effects on human oral cancer cells by modulating cathepsin L expression, MAPK pathway and epithelial-mesenchymal transition" by Velmurugan et al. deals with a very hot topic - advanced stage oral cancers. Their study focuses on the activity of platyphyllenone towards oral squamous carcinomas - FaDu, Ca9-22 and HSC3 - a drug active towards lung and breast cancers. Authors found that the investigated compound does not exhibit antiproliferative activity, however it exerts antimetastatic effect on the studied cell lines through modulation of the expression of L-cathepsin, MAPK pathway and epithelial-mesenchymal transition.

  1. I find the research plan being correctly planned and presented. Despite the poor antiproliferative activity of the compounds in the tested range of concentrations Authors decided to perform more detailed study on the mechanism of action in cells. Anyway, I wonder if the antiproliferative effect would be visible in the higher concentrations of drug as in case of lung and breast cancers?

Answers: Thanks for this valuable suggestion. Indeed, according to our published article, platyphyllenone (40 mM) can induce both autophagy and apoptosis in oral cancers. Platyphyllenone can antiproliferative in the higher concentrations as in case of lung and breast cancers. However, our aim is to test the antimigration effect of drug using lower concentration (20 mM). The reason of inhibit proliferation by drugs can be ruled out in this study.

  1. I think that the time range in motility studies 6 h vs. 24 h is quite long, since the observed changes in most cases are significant. Did Authors do more pictures between these time intervals?

Answers: We apologize for these confuse and thanks for this valuable suggestion. In our previously published article, the observed time point for wound healing analysis was according to the cell movement speed. Therefore, we chose 3, 6 and 24 h to observe the cell motility. We apologize no more pictures between these time intervals.

  1. I think that the introduction lacks the information concerning platyphyllenone - what is the structure?

Answers: We apologize for these confuse and thanks for this valuable suggestion. We have added the chemical structure of platyphyllenone in new figure 1 and follow sentence has been added and highlight with red in revised manuscript.

Line 266: Platyphyllenone (Figure 1A) with 98% purity was purchased from ChemFaces (CheCheng Rd, Wuhan, PRC).

  1. Why did the Authors decide to check its activity towards oral cancers?
    Answers: We apologize for these confuse and thanks for this valuable suggestion. Oral cancer is the one of leading cause of cancer death in Taiwan. Oral cancer treatments are very expensive and often associated with severe side-effects. Although treatment of early-stage oral cancers can be effectively done by surgical re-section along with post-operative chemotherapy and/or radiation therapy, advanced stage oral cancers with lymph node metastasis can be lethal and are associated with poor prognosis. Therefore, we decide to focus on oral cancer rather than other cancer cells.

  1. line 136 did not instead of "didn't"

Answers: We apologize for these confuse. The mistakes has been proofed and highlight with red in revised manuscript.